# Transcriptome-wide analysis of the differences between MCF7 cells cultured in DMEM or αMEM

Yang Jiao[1,2☯], Hongbo Zhao[3☯], Lin Lu[2☯], Xiangyu Zhao[4], Yanchun Wang[2], Bingrong Zheng[5]*

1 NHC Key Laboratory of Periconception Health Birth in Western China, Kunming, 650500, Yunnan, China,
2 Biomedical Engineering Research Institute, Kunming Medical University, Kunming, Yunnan, China,
3 Department of Laboratory Animal Science, Kunming Medical University, Kunming, Yunnan, China,
4 Wuhuajianmei Dental Clinic, Kunming, Yunnan, China, 5 School of Medicine, Yunnan University, Kunming, Yunnan, China

☯ These authors contributed equally to this work.
* zhengbr@ynu.edu.cn

**Data Availability Statement:** The mRNA-seq data from this study has been deposited in the NCBI sequence read archive under the BioProject number PRJNA779251, with the fq files spanning accession numbers SRR16914480-SRR16914481.

## Abstract

MCF7 cells have been used as an experimental model for breast cancer for decades. Typically, a culture medium is designed to supply cells with the nutrients essential for their continuous proliferation. Each medium has a specific nutritional composition. Therefore, cells cultured in different media may exhibit differences in their metabolism. However, only a few studies have investigated the effects of media on cells. In this study, we compared the effects of Dulbecco's modified Eagle medium (DMEM) and minimum essential medium alpha modification (αMEM) on MCF7 cells. The two media differentially affected the morphology, cell cycle, and proliferation of MCF7 cells, but had no effect on cell death. Replacement of DMEM with αMEM led to a decrease in ATP production and an increase in reactive oxygen species production, but did not affect the cell viability. RNA-sequencing and bioinformatic analyses revealed 721 significantly upregulated and 1247 downregulated genes in cells cultured in αMEM for 48 h compared with that in cells cultured in DMEM. The enriched gene ontology terms were related to mitosis and cell proliferation. Kyoto encyclopedia of genes and genomes analysis revealed cell cycle and DNA replication as the top two significant pathways. MCF7 cells were hypoxic when cultured in αMEM. These results show that the culture medium considerably affects cultured cells. Thus, the stability of the culture system in a study is very important to obtain reliable results.

## Introduction

Breast cancer is the most prevalent cancer in women. With a global incidence of 24.2%, breast cancer ranked first among cancer types in women in 2018 [1]. Basic cancer research has mainly used cell lines as the main experimental model. Cultured cells have led to major discoveries in various fields of cancer research, including those related to the occurrence and

Direct link to data: https://www.ncbi.nlm.nih.gov/bioproject/PRJNA779251/ The data supporting the research results can also be obtained from the corresponding authors according to reasonable requirements.

**Funding:** This work was supported by Major Project of Yunnan Science and Technology Program [grant number 202002AA100007 and 202102AA100007-3], Scientific Research Foundation of the Education Department of Yunnan Province [grant number 2021J0226], Joint Special Funds for the Department of Science and Technology of Yunnan Province-Kunming Medical University [grant numbers 2019FE001(-176), 202101AY070001-075]. The amended role of Funder statement are as follows: BR.Z. Grant number 202002AA100007. Major Project of Yunnan Science and Technology Program. Study design. https://kjt.yn.gov.cn BR.Z. Grant number 202102AA100007-3. Major Project of Yunnan Science and Technology Program. Decision to publish. https://kjt.yn.gov.cn BR.Z. Grant number 2021J0226. Scientific Research Foundation of the Education Department of Yunnan Province. Preparation of the manuscript. https://jyt.yn.gov.cn HB.Z. Grant numbers 202101AY070001-075. Joint Special Funds for the Department of Science and Technology of Yunnan Province-Kunming Medical University. Data collection and analysis. https://kjt.yn.gov.cn L.L. Grant numbers 2019FE001(-176). Joint Special Funds for the Department of Science and Technology of Yunnan Province-Kunming Medical University. Data collection and analysis. https://kjt.yn.gov.cn.

**Competing interests:** The authors have declared that no competing interests exist.

development of cancers [2], anticancer treatment [3], and prognosis [4]. Approximately 84 human breast cancer cell lines have been classified by molecular subtyping into five subtypes: luminal A, luminal B, HER-2$^+$, triple$^-$ A, and triple$^-$ B [5]. Each cell line is cultured in a standard medium. MCF7 cells are of the luminal A subtype and are cultured in Eagle's minimum essential medium (EMEM) with 0.01 mg/mL human recombinant insulin and 10% fetal bovine serum (FBS) according to the guidelines of the American Type Culture Collection (ATCC) [6]. However, in some laboratories, they are cultured in Dulbecco's modified Eagle medium (DMEM) containing 10% FBS [7–11]. In our laboratory, we have been culturing this cell line in DMEM since we received it (at least 15 years ago) from a laboratory where this cell line was also cultured in DMEM. Therefore, we considered DMEM as the usual medium for MCF7 cells. The culture medium should not be changed during an experiment, especially when seeding cells, as the cells would otherwise be subjected to stress resulting from nutritional changes [12]. Cells may finally adapt to the new environment over time, but their metabolic and gene expression profiles are likely to be modified during this period. For example, when breast cancer cell lines were separately cultured in three culture media for four passages, their phenotypic differentiation markers differed and the expression of these markers depended on the composition of the medium [13]. When cancer cells are cultured with excessive concentrations of pyruvate, their proliferation depends less on mitochondrial respiration compared with that of cells cultured with a regular concentration of pyruvate [14]. Therefore, we wanted to check if temporarily replacing DMEM with other media would affect the cells.

In conventional cell culture, the culture medium is designed to supply cells with the nutrients essential for continuous proliferation [15]. Currently, DMEM and minimum essential medium alpha modification (αMEM) are the widely used cell culture media. Each medium has a specific formulation. DMEM is a universally used culture medium for various primary cell types and immortal cell lines. αMEM is enriched with non-essential amino acids and additional vitamins [16]. In this study, we cultured MCF7 cells in DMEM or αMEM and compared their gene expression and metabolic profiles. We aimed to comprehensively investigate the effects of the culture medium stability on cultured cells.

## Materials and methods

### Cell culture

The breast cancer cell line MCF7 used in this study was obtained from the Kunming Cell Bank of the Chinese Academy of Sciences. It was maintained in DMEM supplemented with 10% FBS without heat inactivation (10270–106, Gibco, Life Technologies™, Australia), 100 mg/L penicillin, and 100 mg/L streptomycin (15140–122, Gibco, Life Technologies™, NY, USA) at 37˚C with 5% $CO_2$. DMEM contains 4.5 g/L glucose, L-glutamine, and 110 mg/L sodium pyruvate (C11995500BT; Gibco, Thermo Fisher Scientific, Suzhou, China).

### Exposure of MCF7 cells to a different medium

MCF7 cells were seeded at a density of $1 \times 10^5$ cells/mL per well in a 6-well plate (3516, Corning, NY, USA) and cultured overnight in DMEM. Subsequently, the medium in the treatment group was replaced with 2 mL of αMEM, supplemented with 10% FBS without heat inactivation, 100 mg/L penicillin, and 100 mg/L streptomycin (Gibco by Life Technologies™, NY, USA). αMEM contained L-glutamine, Ribonucleosides and Deoxyribonucleosides (SH30265.01; Hyclone, Thermo Scientific, Beijing, China). The medium in the control group was replaced with 2 mL of fresh DMEM.

## Flow cytometric analysis of the cell cycle

Control MCF7 cells and those cultured in αMEM for 48 h were collected after digestion with trypsin without EDTA (T1350, Solarbio, Beijing, China) and fixed overnight in ice-cold 70% ethanol. The fixed cells were stained with 10 μL propidium iodide (PI, CW2574S, CWBIAO, Beijing, China) in phosphate-buffered saline (PBS) containing 0.1% Triton X-100 (T8200, Solarbio, Beijing, China) and 50 μg/mL RNase A (CW0601, Beijing, China) and analyzed using the CyFlow Space (Partec) and FolMax 2.82 software.

## Annexin V-FITC/PI staining

The percentage of apoptotic cells was determined using an Annexin V-FITC/PI Apoptosis Detection Kit (CW2574S, CWBIO, Beijing, China). Briefly, control MCF7 cells and those cultured in αMEM for 48 h were collected after digestion with trypsin without EDTA and then washed with cold PBS and resuspended in the binding buffer at a concentration of $10^6$ cells/ mL. The cells were incubated at 37°C with 5 μL of Annexin V-fluorescein isothiocyanate (FITC) and 10 μL of PI for 10 min and analyzed using the CyFlow Space (Partec) and FolMax 2.82 software.

## Cell attachment assay

MCF7 cells were digested with trypsin and centrifuged in two tubes. The cell pellet was resuspended separately in DMEM and αMEM, and $1 \times 10^4$ cells were seeded per well in a 96-well plate after counting. The wells were washed with PBS to remove nonadherent cells, and then 100 μL of fresh medium and 20 μL of CellTiter 96 Aqueous One Solution Reagent were added to detect cell viability at 15, 30, 45, 60, and 75 min.

## Colony formation assay

Cells in logarithmic growth phase were collected after trypsin digestion, resuspended in culture medium, and counted. These cells were inoculated in a six-well plate (1000 cells per well), and 2 mL αMEM or DMEM was added to the treatment and control groups, respectively. The medium was refreshed every three days, and the cells were cultured for 3 weeks when the majority of colonies had more than 10 cells. The cells were washed once with PBS. One milliliter of 4% paraformaldehyde was added to each well and the cells were fixed for 30 min. The cells were washed once with PBS and then stained with 1 mL of crystal violet added to each well for 15 min. After washing the cells three times with PBS, they were air-dried and photographed, and the number of clones was counted. The colony formation rate was calculated using the following equation:

Clone formation rate = (number of clones/number of inoculated cells) ×100%

## ATP detection

The Mitochondrial ToxGlo$^{TM}$ Assay (Promega, Madison, WI, USA) was used to detect intracellular ATP levels in MCF7 cells. The experiment was performed following the steps prescribed in the technical manual provided by the manufacturer. Briefly, MCF7 cells were seeded at a density of $5 \times 10^3$ cells/well in a 96-well plate (3599, Corning) containing 100 μL DMEM and cultured overnight. Subsequently, the medium was replaced with 100 μL αMEM or DMEM for the treatment and control groups, respectively. At the detection time points, the cells were washed with glucose-free culture medium twice, and then 80 μL glucose-free culture medium and 20 μL of 5× Cytotoxicity Reagent were added to each well. The plates were

incubated at 37˚C for 30 min, and then at 25˚C for 5 min. Subsequently, 100 μL ATP Detection Reagent was added to each well and the luminescence generated was measured.

## ROS and cell proliferation assays

The ROS-Glo™ $H_2O_2$ Assay (Promega) was used to detect the levels of reactive oxygen species (ROS). The experiment was performed following the steps prescribed in the technical manual provided by the manufacturer. Briefly, MCF7 cells were seeded at a density of $5\times10^3$ cells/well in a 96-well plate (3599, Corning) containing 100 μL DMEM and cultured overnight. Subsequently, the medium was replaced with 80 μL αMEM for the treatment group or with 80 μL fresh DMEM for the control group. The $H_2O_2$ substrate solution (20 μL) was added to the wells at 0, 18, and 42 h, and plates were further incubated at 37˚C with 5% $CO_2$ for 6 h. The entire medium (100 μL) in the wells was transferred to a new well, and 100 μL of ROS-Glo™ Detection Solution was added to each well. The plate was incubated for 20 min at 25˚C and subsequently relative luminescence units were recorded. Simultaneously, 100 μL fresh medium was added to the original assay well along with 20 μL of CellTiter 96 Aqueous One Solution Reagent, and the plate was incubated at 37˚C in a 5% $CO_2$ incubator. After 3 h, the absorbance of each sample was recorded at 490 nm.

## RNA isolation and RNA-Seq library preparation

Control MCF7 cells and those cultured in αMEM ($10^6$ cells/mL) for 48 h were harvested (three biological replicates per sample), and their total RNA was extracted using TRIzol™ reagent (93289, Invitrogen, Carlsbad, USA). The RNA quality and quantity were determined using the RNA Nano 6000 Kit in the Agilent 2100 Bioanalyzer System, and the samples were stored at –80˚C until needed. For construction of cDNA libraries, 10 μg total RNA per sample was used. A total of six samples were sequenced using the BGISEQ-500 platform.

On average, approximately 23.94 Mb clean reads (S1 Table) were generated per sample, and 18,174 genes were detected. The average mapping ratio of clean reads to the reference human genome was 94.82% and the uniformity of the mapping results per sample suggested that the two groups were comparable. Raw reads were filtered using the internal software, SOAPnuke, to obtain clean reads [17]. The clean reads were mapped to reference transcripts (Human hg19) using Bowtie2 [18].

## Differential gene expression and gene enrichment analysis

Gene expression was expressed as the number of uniquely mapped reads per kilobase of exon fragments per million mappable reads (FPKM) with RSEM [19]. To reflect the gene expression correlation between samples, we calculated the Pearson correlation coefficients for all gene expression levels between each pair of samples and reflected these coefficients in the form of heat maps, as shown in S1 Fig. We then performed a principal components analysis (PCA) based on the PCA plan, as shown in S2 Fig.

Based on the gene expression levels, the differentially expressed genes (DEGs) were identified using the DEGseq algorithm [20]. For the analysis, a $q$-value $\leq0.001$ and an absolute value of $log_2$ ratio $\geq1$ were used to verify the significance of differences in gene expression between the two samples. Based on the DEGs, pathway function analysis was performed using the clusterProfiler package [21]. Enrichment analysis was carried out using the hypergeometric test with a threshold value of 0.05 based on the Kyoto encyclopedia of genes and genomes (KEGG) database.

## Validation of sequencing-based gene expression levels via quantitative real-time polymerase chain reactions (qPCR)

Total RNA from control MCF7 cells and those cultured in αMEM for 48 h was extracted using the TRIzol™ reagent (93289, Invitrogen). The quality and quantity of the RNA samples was determined using a Nano-300 spectrophotometer (Allsheng). The samples were used to synthesize the first-stand cDNA using the PrimeScript™ RT Reagent Kit with gDNA Eraser according to the manufacturer's instructions (RR047A, Takara, China). The primer sequences (Table 1) were designed using PrimerQuest (http://www.idtdna.com/primerquest/Home/Index). qPCR was performed using the BIO-RAD CFX96™ Real-Time System and FastStart Universal SYBR Green Master (06924204001, Roche, Mannheim, Germany). Relative quantification was performed using the comparative Ct ($2^{-\triangle\triangle Ct}$) method.

## Western blot analysis

MCF7 cells cultured in DMEM or αMEM for 48 h were lysed in 50 μL RIPA lysis buffer (strong) (P0013B, Beyotime, Shanghai, China) containing 0.5 μL phenylmethylsulphonyl fluoride (ST506, Beyotime), and the lysates were incubated on ice for 30 min. The lysates were then incubated with 50 μL of 2X SDS-PAGE protein loading buffer (Bio-Rad) in boiling water for 10 min. The supernatant was subjected to electrophoresis on a 10% sodium dodecyl sulfate-polyacrylamide gel and the resolved proteins were transferred onto polyvinylidene difluoride membranes (IPVH00010, Merck Millipore). The membranes were blocked with TBST (0.01 M Tris-buffered saline (TBS) with 0.1% Tween-20, pH 7.4) containing 5% non-fat dried milk for 1 h and incubated overnight at 4˚C with antibodies against GAPDH (RRID: AB_2801390, CW0100M, 1:2000; CWBio, Jiangsu, China), P21 (RRID: AB_10860537, ab109520, 1:500, abcam, Shanghai, China), CDK1 (RRID: AB_11218160, AM06438SU-N, 1:100; Origene, Wuxi, China), and β-actin (RRID: AB_2665433, CW0096M, 1:1000; CWBio, Jiangsu, China). The membranes were then incubated with horseradish peroxidase-conjugated goat anti-mouse IgG (RRID: AB_2736997, CW0102S, CWBio, Jiangsu, China) or goat anti-rabbit IgG (RRID: AB_2814709, CW0103, 1:2000; CWBio, Jiangsu, China) at 20–25˚C for 1 h and then with the ECL substrate solution (CW0049S, 1:1 [v/v]; CWBio, Jiangsu, China). The bands were got from BioRed gel imaging system (Model No. PowerPac Universal Power Supply) with Image Lab 5.2 system. The protein bands quantified using Photoshop 7.0.

**Table 1. Sequences of the primers used in the quantitative real-time polymerase chain reactions.**

| Primers | NCBI ID | Primer sequences |
|---------|---------|------------------|
| P21 | NM_000389 | S:5′ TTAGCAGCGGAACAAGGAGTCAGA3′<br>A:5′ ACACTAAGCACTTCAGTGCCTCCA 3′ |
| CDK1 | NM_001170406.1 | S:5′GCCTGAGATAACATAGAACTGGTAG3′<br>A: 5′CTCTGCCCTAGGCTTTCATTAC 3′ |
| GAPDH | NM_002046.7 | S: 5′GGTGTGAACCATGAGAAGTATGA 3′<br>A: 5′GAGTCCTTCCACGATACCAAAG 3′ |
| BIRC3 | NM_001165.4 | S: 5′ CAAGCCAGTTACCCTCATCTAC 3′<br>A: 5′ CTGAATGGTCTTCTCCAGGTTC 3′ |
| ITGA1 | NM_181501.2 | S: 5′ GACTGGCTTCAGTGCTCATTA 3′<br>A: 5′ TTGACTAGCCTTCTGCATGAC 3′ |
| β-actin | NM_001101.5 | S: 5′ TGACGTGGACATCCGCAAAG 3′<br>A: 5′ CTGGAAGGTGGACAGCGAGG 3′ |

### Statistical analyses

Statistical analyses were performed using Microsoft Excel 2007 and R 4.2.3. Graphs were drawn using the GraphPad Prism 5 software and R 4.2.3. A value of $P < 0.05$ was considered to indicate a statistically significant difference.

## Results

### Effect of culture medium on cell cycle and proliferation

To study the effect of culture medium on cells, we cultured MCF7 cells in DMEM or αMEM with 10% FBS. MCF7 cells cultured in DMEM displayed the characteristic "cobblestone" morphology with a clear boundary and were tightly packed and flat. However, the cells cultured in αMEM for 48 h displayed groups of irregular cells showing cell-to-cell connections (Fig 1; marked with red arrow). We evaluated the cell cycle distribution of MCF7 cells cultured in DMEM and αMEM for 24 and 48 h using PI staining. No obvious changes were noted at 24 h; however, at 48 h, the number of cells in the G0/G1 and G2/M phases decreased and increased significantly, respectively (Fig 2). As evident from the results of Annexin V/PI staining, there was no significant difference in cell death between the two groups (Fig 3). Next, we measured the ability of cells to attach to the surface of the cell culture flasks in the two culture media. The MCF7 cells in DMEM adhered to the flask surface more rapidly (45 min) than they did in αMEM (60 min) (Fig 4). The ability to adhere to the flask surface reflects the survival of the cells after seeding. However, not every adherent cell proliferates or forms a clone. Formation of colonies requires both cell adhesion and proliferation. Therefore, we tested the colony-forming ability of the cells. The rate of colony formation in MCF7 cells cultured in DMEM was 40.07 ± 3.63% (Fig 5). However, in αMEM, although cell adhesion was observed 24 h after seeding, the cells gradually died upon culture and there were no surviving cells at the time of harvesting. This indicated that the proliferation activity of MCF7 cells was affected in αMEM.

### Effect of culture medium on ATP and ROS production

ATP is the most important energy molecule in cells and is a key substance for maintaining normal cellular activity. In standard glucose-containing culture media, propagated cells tend

| DMEM | αMEM |
|---|---|

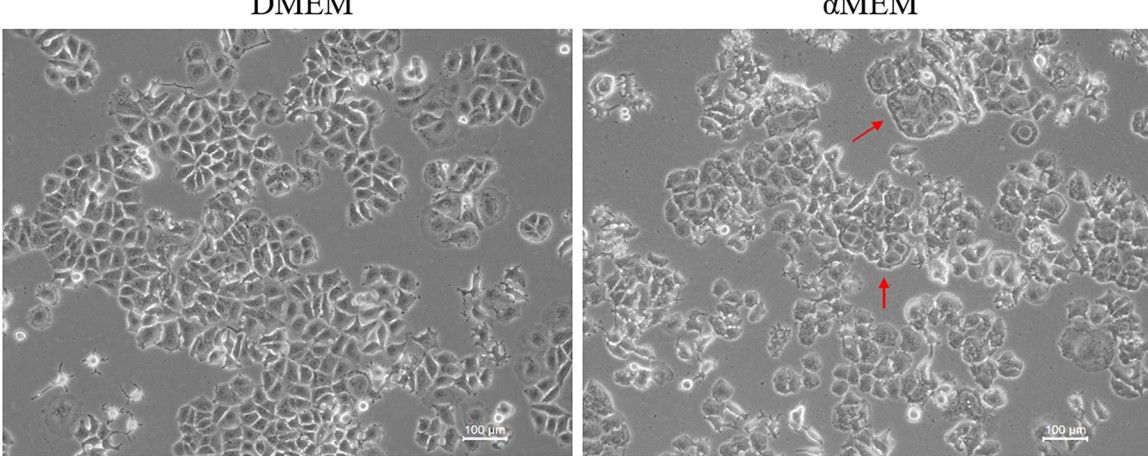

**Fig 1. Cell morphology.** Images were obtained after 48 h of culture in DMEM or αMEM. Representative micrographs from at least three independent experiments are shown. All the images were acquired at 20× magnification.

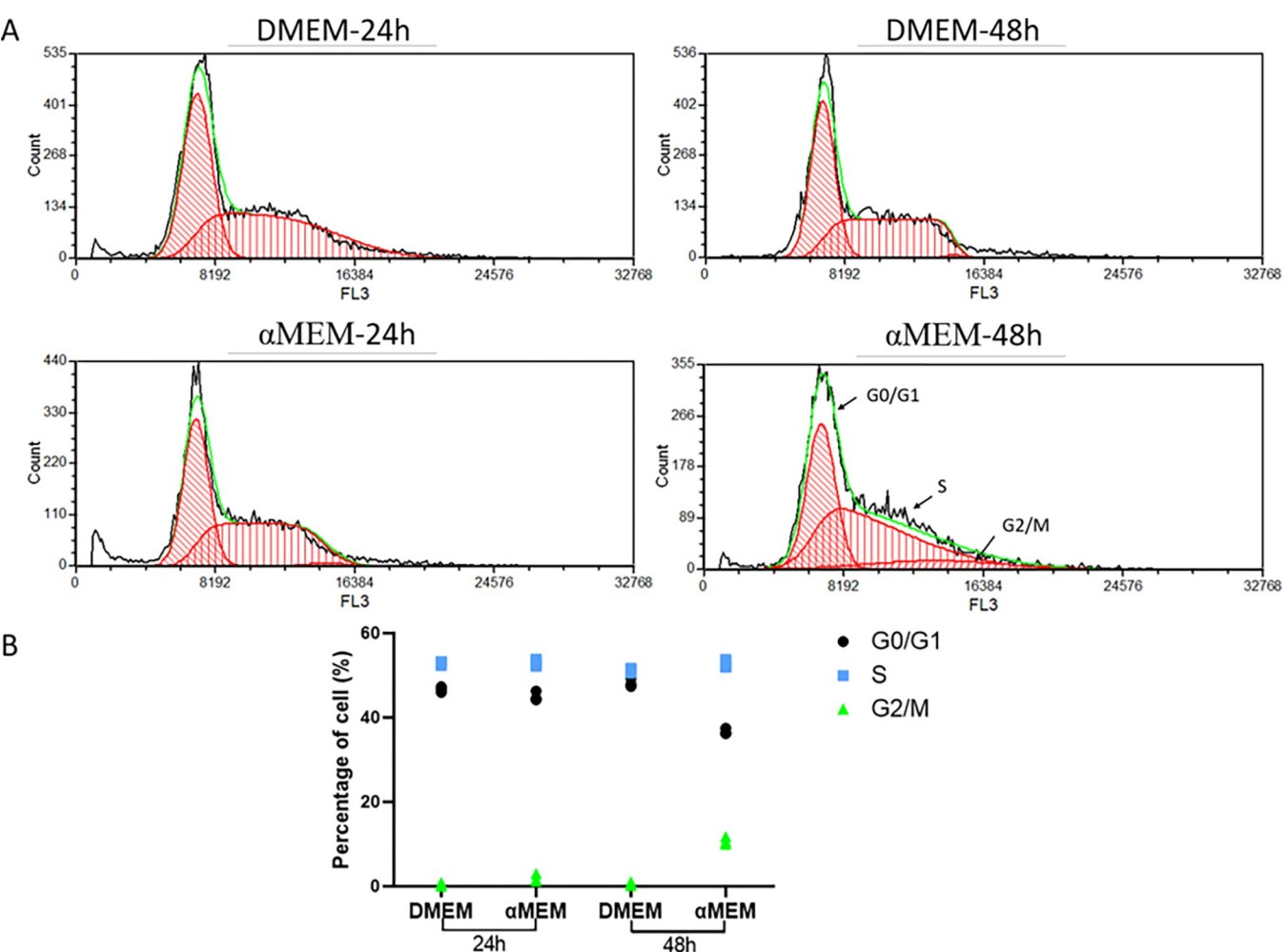

**Fig 2. Cell cycle analysis.** A. Flow cytometry results. Cells were collected after 24 h or 48 h cultured in DMEM or αMEM and stained with PI. B. Statistical chart of cell cycle. Data were presented as the mean ± S.D. from three independent experiments.

to use glycolysis to meet their ATP needs. We investigated whether a change in the culture medium would lead to changes in intracellular ATP levels, thereby, affecting cell growth. Replacement of DMEM (high sugar) with αMEM led to a decrease in ATP production in MCF7 cells (Fig 6A). Simultaneously, ROS production continued to increase (Fig 6B). However, during this process, there was no significant difference in cell viability (Fig 6C).

## Identification of DEGs in cells cultured in different media

According to the results of RNA-seq (The mRNA-seq data from this study has been deposited in the NCBI sequence read archive under the BioProject number PRJNA779251 (https://www.ncbi.nlm.nih.gov/bioproject/PRJNA779251/), with the fq files spanning accession numbers SRR16914480-SRR16914481) and bioinformatics analysis, a total of 721 up- and 1247 downregulated genes were identified [|log2 (fold change) | >1, q-value ≤0.001] (Fig 7).

Next, we performed GO classification and functional enrichment analyses of the DEGs. The DEGs were assigned three ontologies: biological processes (BP), cellular component (CC), and molecular biological function (MF). The top-10 enriched terms for each ontology are shown in Fig 8. All the terms were closely related to mitosis, consistent with the proliferating

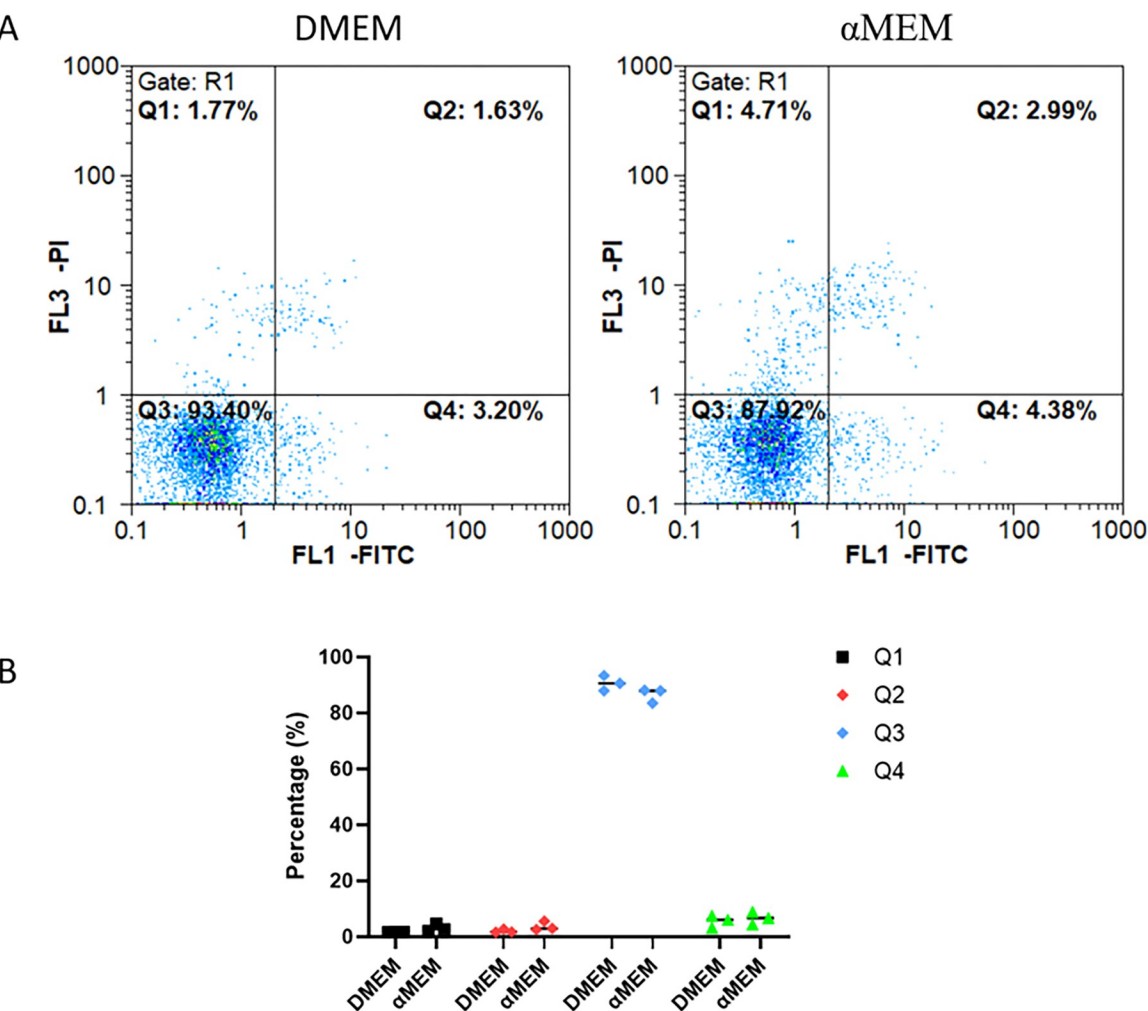

**Fig 3. Flow cytometry results of Annexin V-FITC/PI staining.** A. Apoptotic status. MCF7 cells were collected after 48 h cultured in DMEM or αMEM. The cultured cells were stained with 5 μL Annexin V-FITC and 10 μL PI. The cells in Q1, Q2, Q3, and Q4 quadrants are necrotic, late-apoptotic, viable cells, and early-apoptotic, respectively. B. Quantitative analysis for cell apoptosis rate. The results are presented as the mean ± S.D. from three independent experiments.

state of the cells. The culture medium affected cell cycle and proliferation, consistent with the results of cell viability and cell cycle analyses.

To identify the biological functions associated with the DEGs, we performed KEGG pathway analysis (Fig 9), which revealed 19 significant pathways ($p < 0.05$). "Cell cycle" and "DNA replication" were the top-two terms, which confirmed that the culture medium affected the cell cycle and proliferation of MCF7 cells.

## Validation of the sequencing results by determining the gene expression levels

Among the DEGs, *P21*, *GAPDH*, and *BIRC3*, were significantly upregulated, whereas *CDK1* and *ITGA1* were significantly downregulated (Table 2). To validate these gene expression patterns, we assessed their mRNA levels using qPCR and observed that the qPCR results were consistent with the RNA-Seq results (Fig 10).

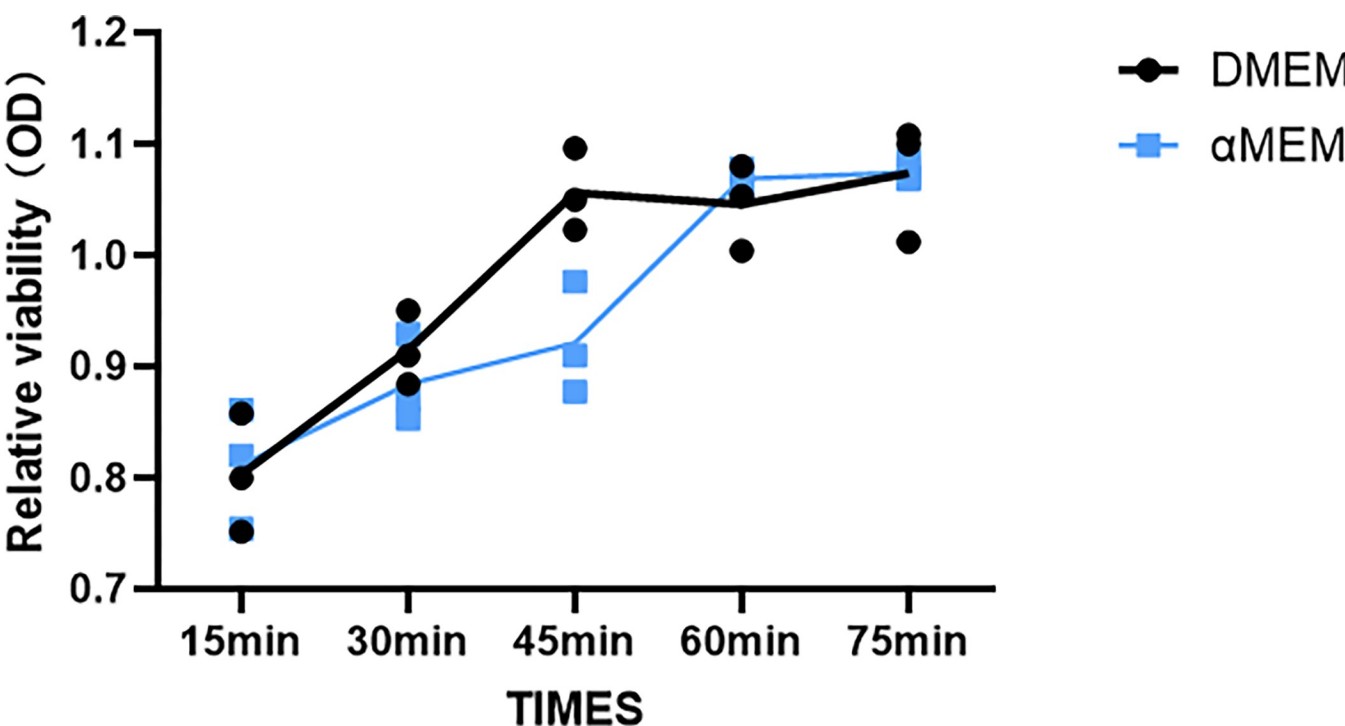

**Fig 4. The detection of cell attachment to the surface of the cell culture flasks.** The X-axis represents the cell adhesion time. The Y-axis represents cell viability.

To evaluate the levels of protein encoded by the DEGs, we performed western blot analysis of proteins extracted from MCF7 cells cultured in DMEM or αMEM. The P21 and GAPDH levels were significantly increased, whereas CDK1 levels were significantly decreased at 24 h. At 48 h, the P21 and CDK1 levels showed no obvious changes, but GAPDH levels were increased (Fig 11).

## Discussion

Each cell line required a specific medium for its optimal culture. MCF7 cells are generally cultured in DMEM supplemented with 10% FBS [7–10]. To evaluate the effect of culture medium on cells, we cultured MCF7 cells in two different media (DMEM and αMEM supplemented with 10% FBS). After 48 h of culture, the cells cultured in αMEM exhibited a different morphology, cell cycle, and proliferation compared with those cultured in DMEM; however, there was no significant difference in cell death. Upon replacement of DMEM with αMEM, the ATP production in MCF7 cells decreased, and ROS production increased but there was no significant difference in cell viability. To evaluate the changes in gene expression, we performed RNA-Seq analysis of MCF7 cells cultured in each medium. A total of 721 genes were significantly upregulated and 1247 genes were significantly downregulated in MCF7 cells cultured in αMEM compared with those cultured in DMEM. These results indicate that the medium change had a significant effect on the growth of MCF7 cells. Therefore, stable culture conditions are crucial for the stability and reliability of experimental results, especially for high-throughput sequencing analysis.

To decipher the changes in MCF7 cells caused due to change in the culture medium, we performed RNA-Seq analysis and subjected the data to bioinformatic analysis. The results showed that differential gene enrichment of biological processes was mainly related to DNA

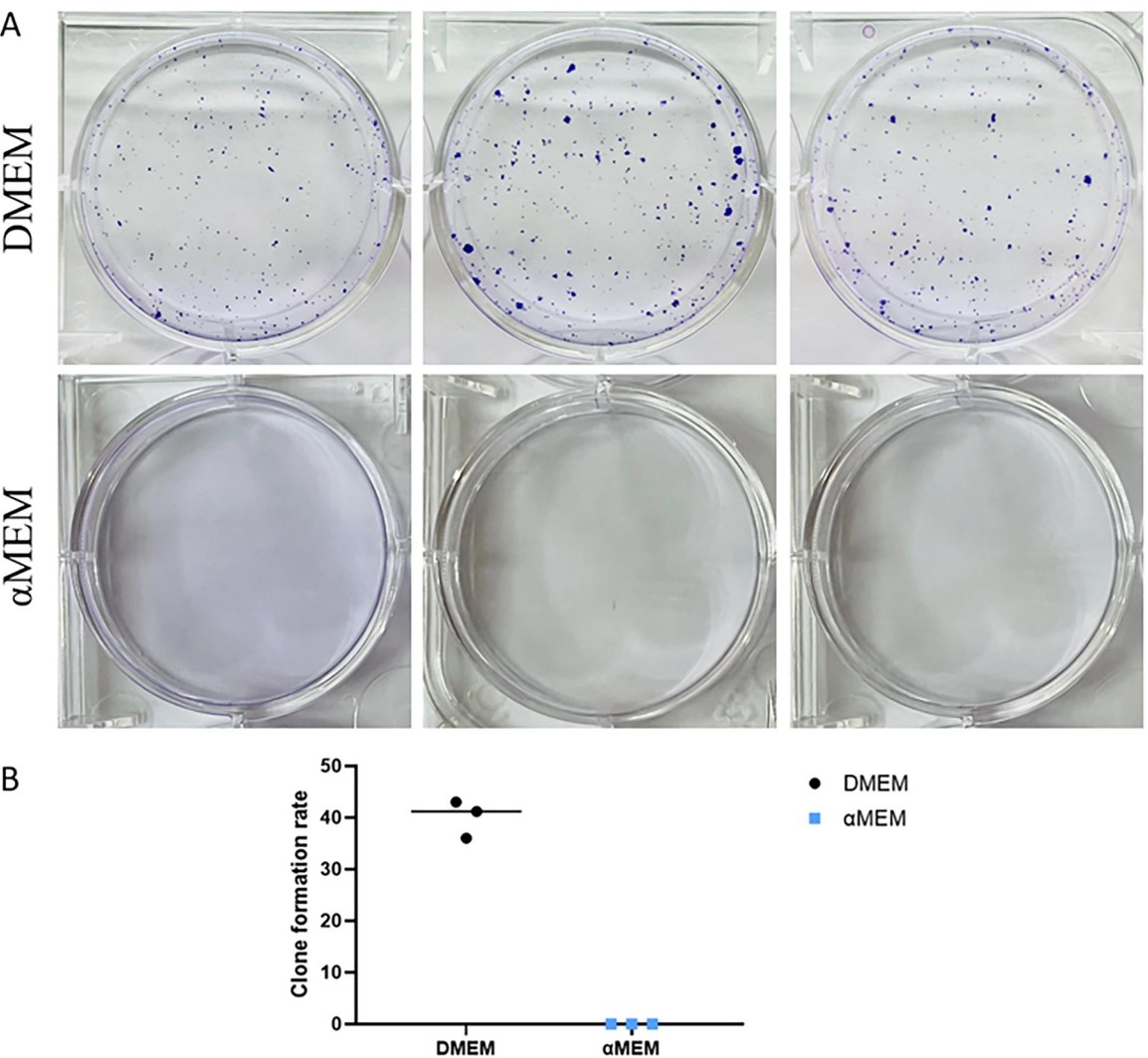

**Fig 5. Colony forming assay.** A. Clonal plaques. Cells were inoculated in a six-well plate (1000 cells per well), and 2 mL αMEM or DMEM was added to the treatment and control groups, respectively. The medium was refreshed every three days, and the cells were cultured for 3 weeks and stained with crystal violet. B. Statistical chart of Colony forming assay. Clone formation rate = (number of clones/number of inoculated cells) ×100%. The results are presented as the mean ± S.D. from three independent experiments.

replication and chromosome segregation. The enrichment of cellular components was mainly related to chromosome concentration, centromere, microtubule, and kinetochore and that of molecular biological functions was mainly related to RNA polymerase II. This enzyme exists in eukaryotic cells and catalyzes the transcription of genes to mRNAs and most hnRNAs and miRNA precursors [22]. All these results suggested that the changes in MCF7 cells cultured in αMEM compared with those cultured in DMEM were related to the G2/M phase of the cell cycle. The G2 phase (second gap phase) encompasses the late stage of DNA synthesis and preparations related to mitosis [23]. During this period, DNA synthesis is terminated, and various genes are expressed, including tubulin and maturation-promoting factors [24]. The M phase is the cell division phase, in which a series of nuclear changes, chromatin condensation, emergence of spindles, and precise and equal distribution of chromosomes into two daughter cells occur [25]. Thus, the results of the GO analysis were consistent with those of the flow cytometric cell cycle analysis.

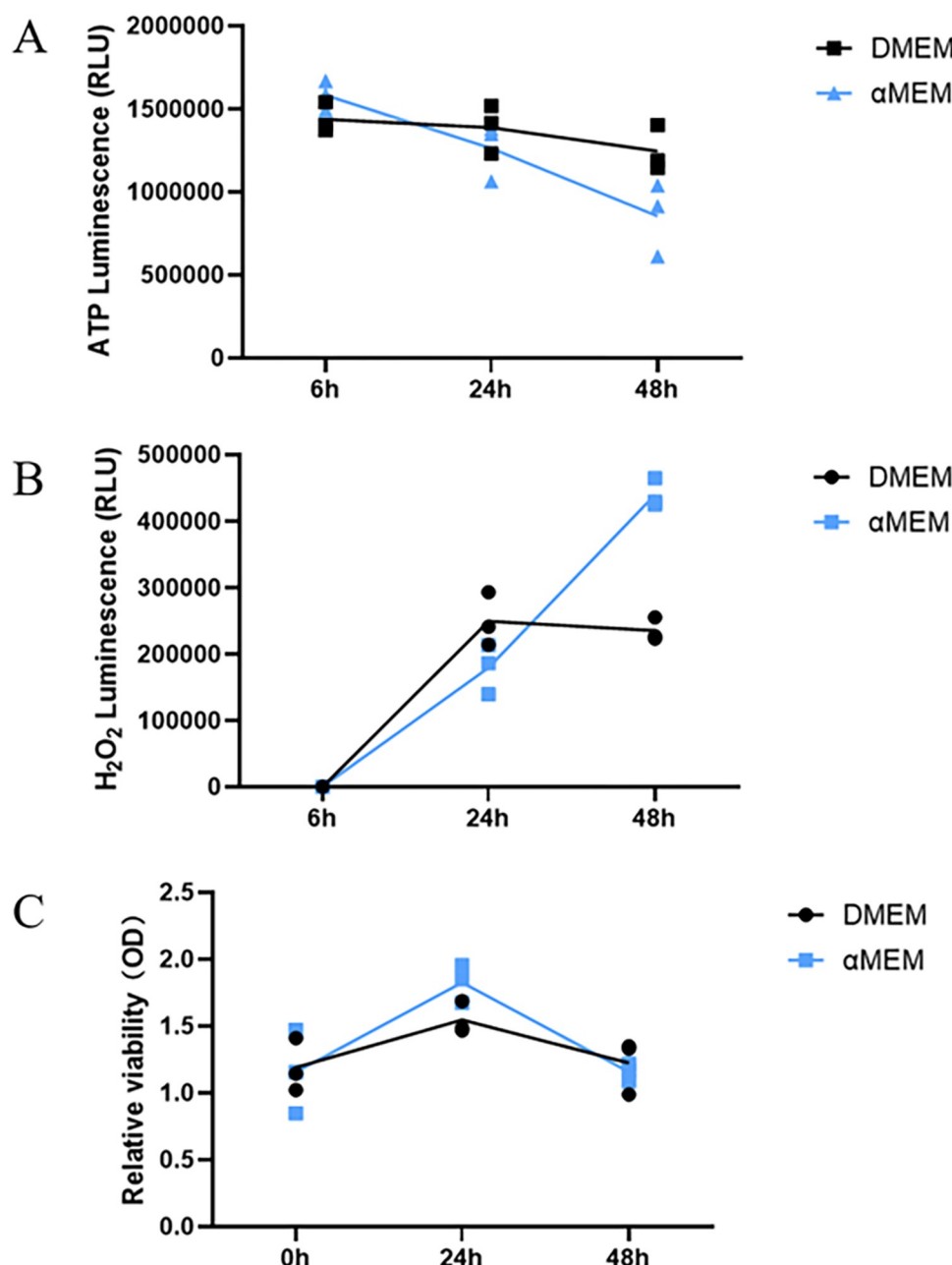

**Fig 6. ATP and ROS production assay.** A. Intracellular ATP levels assay. B. ROS assay. C. Cell viability assay. Intracellular ATP levels and ROS levels were detected at 6 h, 24 h and 48 h after replacement of DMEM with αMEM. At the same time, cell viability was detected. Data were presented of three independent experiments.

According to the KEGG analysis results, the top-two pathways to be affected were cell cycle and DNA replication. This directly shows that the change in the culture medium affects the cell cycle and proliferation considerably. Among the DEGs, P21 (cyclin dependent kinase inhibitor 1A, CDKN1A) [26] was upregulated, and CDK1 (cyclin-dependent kinase 1) was significantly downregulated. P21 can inhibit CDK2 [27], CDK1, and CDK4/6 complexes [28] and, thus, functions as a regulator of cell cycle progression in the G1 and S phases. Additionally, P21 blocks cell cycle progression in the S phase by binding to and inhibiting

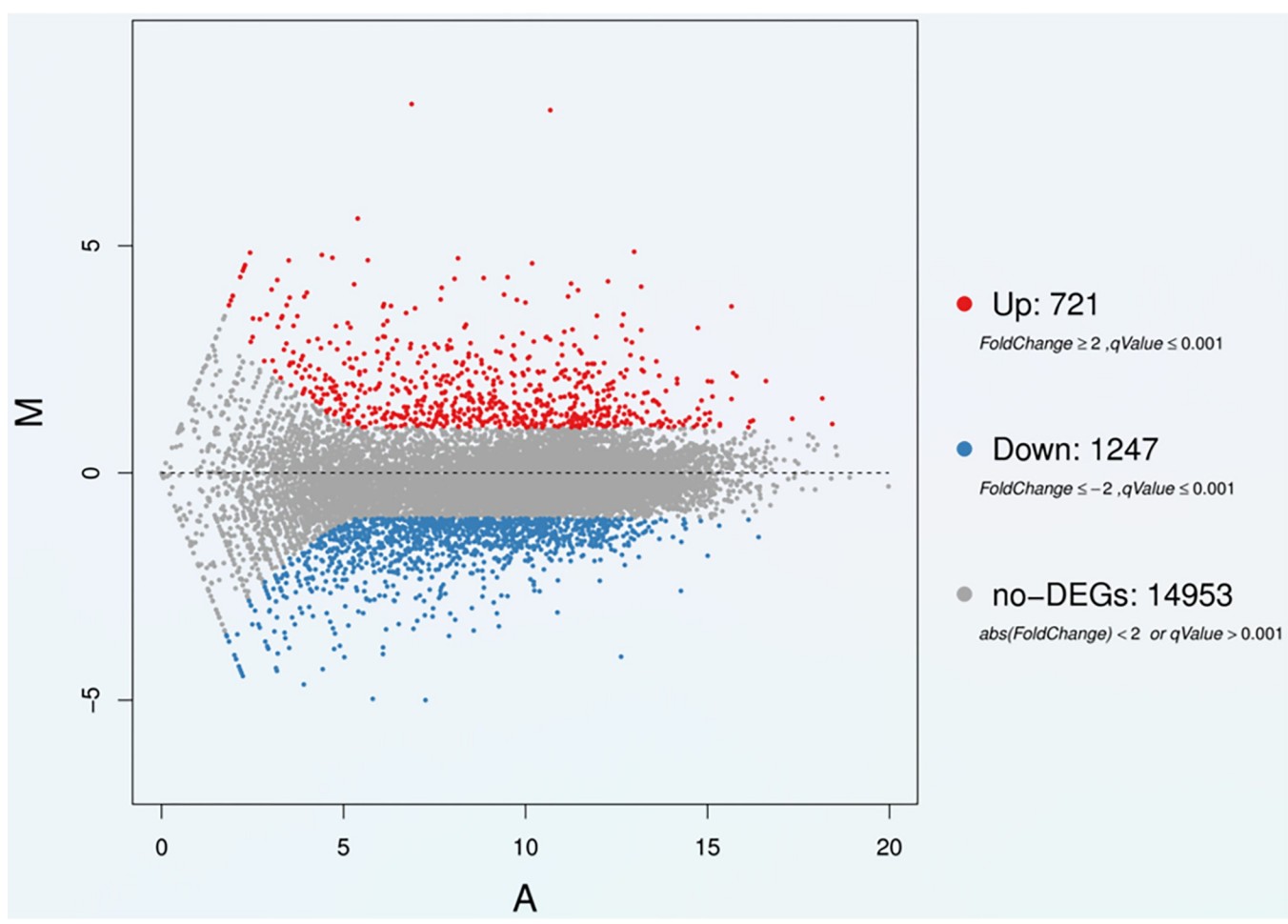

**Fig 7. MA plot of DEGs.** The X-axis represents value A (log2-transformed mean expression level). The Y-axis represents value M (log2-transformed fold change). Red and blue dots represent up- and down-regulated DEGs, respectively. Gray points represent non-DEGs. Cells were collected after 48 h cultured in DMEM or αMEM. n = 3 in each medium.

proliferating cell nuclear antigen (PCNA) [29]. P21 is important for the G2/M transition and mitotic progression [30] and maintains chromosomal integrity in vivo. Loss of P21 completely abolishes the cell cycle arrest function of P53R172P and accelerates the onset of tumorigenesis in Trp53[515C/515C] mice [31]. In fact, P21 is a two-faced genome guardian that not only inhibits, but also activates the cell cycle, depending on its expression levels [28]. Chien-Hsiang Hsu et al. [32] reported the existence of a "Goldilocks zone" for the effect of P21 on cell proliferation. In our study, the number of cells in the G0/G1 phase decreased and that of cells in the G2/M phase increased concomitantly with the upregulation of P21.

CDK1 is required for the transition of cells from the G2 phase to mitosis [33]. The activity of CDK1 is regulated by cyclins and checkpoint kinases, which ensures that the cells do not enter mitosis with incompletely replicated or damaged DNA [34]. CDK1 overexpression correlates with an adverse prognosis in breast cancer [35]. Cells with inhibited CDK1 expression no longer divide but increase in size [36]. In our study, CDK1 was significantly downregulated in cells cultured in αMEM compared with that in cells cultured in DMEM. Additionally, morphological assessment showed that the intercellular boundary was not clear in cells cultured in DMEM, which indicates that the downregulation of CDK1 interfered with

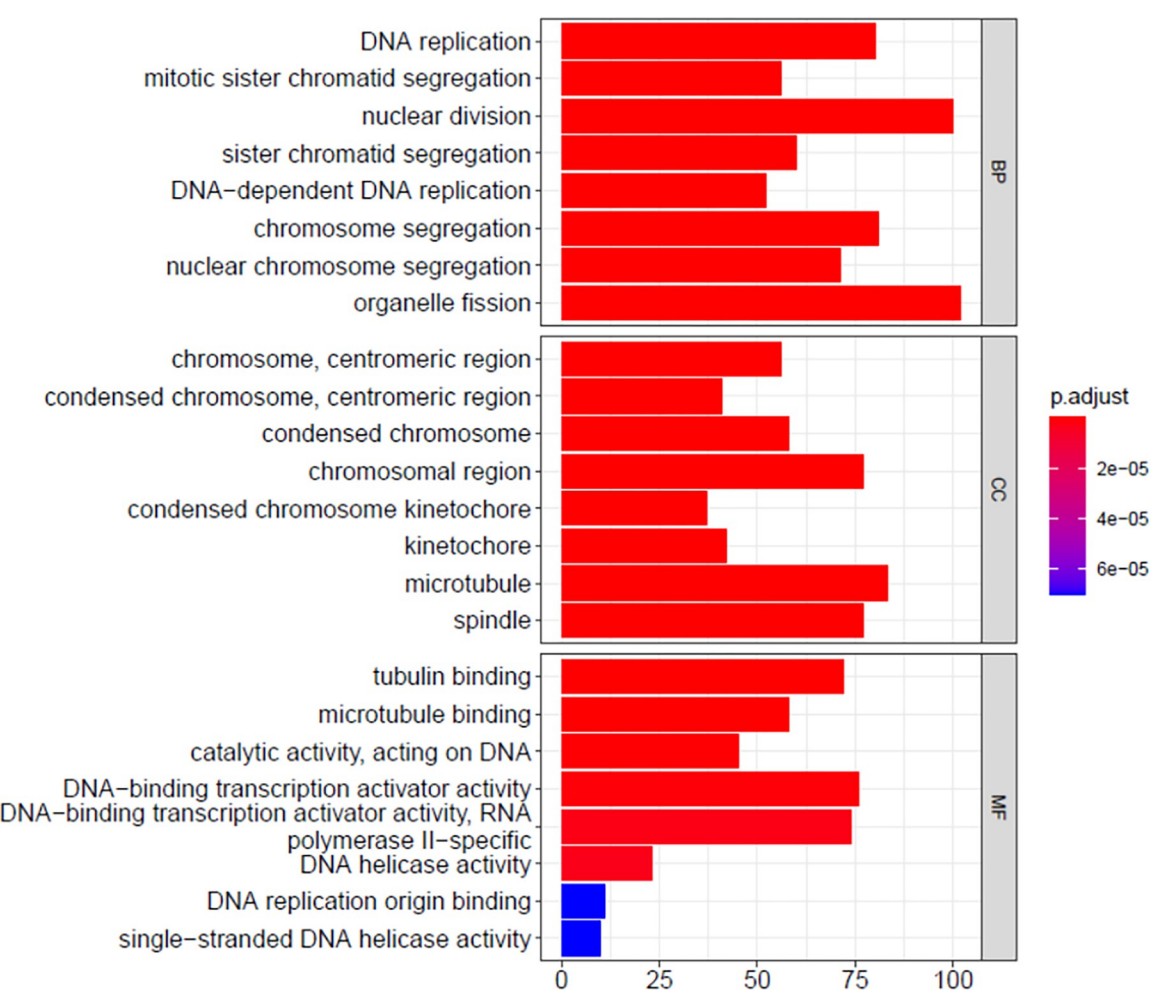

**Fig 8. The gene ontology term enrichment of DEGs.** BP: Biological processes. CC: Cellular components. MF: Molecular function. The X-axis refers to the number of genes in functional analysis. The Y-axis refers to GO terms. The colors, ranging from blue to red, refer to the change in value threshold of the identified significantly enriched GO terms (P-adjust < 0.05) from low to high.

mitosis and cell cycle arrest at the G2/M phase. Recent studies have shown that CDK1 plays a role in regulating the G2/M phase. The activity of CDK1 changes throughout the cell cycle; the CDK1 levels start to increase during the S phase, reach a maximum in the metaphase, and decline rapidly in the anaphase [37]. Katharina et al. [36] reported that CDK1 plays an important role in the translation of 5´ TOP mRNAs, which include mRNAs encoding ribosomal proteins and several translation factors. Thus, CDK1 probably regulates cell proliferation by regulating protein synthesis. Furthermore, the cyclin B1/CDK1 complex mediates mitochondrial activity during cell cycle progression, enabling mitochondria to sense cellular fuel demand and coordinate the ATP output [38]. In our study, along with the significant downregulation of CDK1, we found a significant decrease of intracellular ATP levels in αMEM compared with DMEM. In most tissues, inactivation of CDK1 causes remodeling of integrin-receptor adhesion complexes and the actin cytoskeleton; consequently, cells rapidly enter mitosis [39]. Considering the complex and diverse functions of CDK1, additional research is needed to elucidate its role in mediating the effects of culture medium on cells.

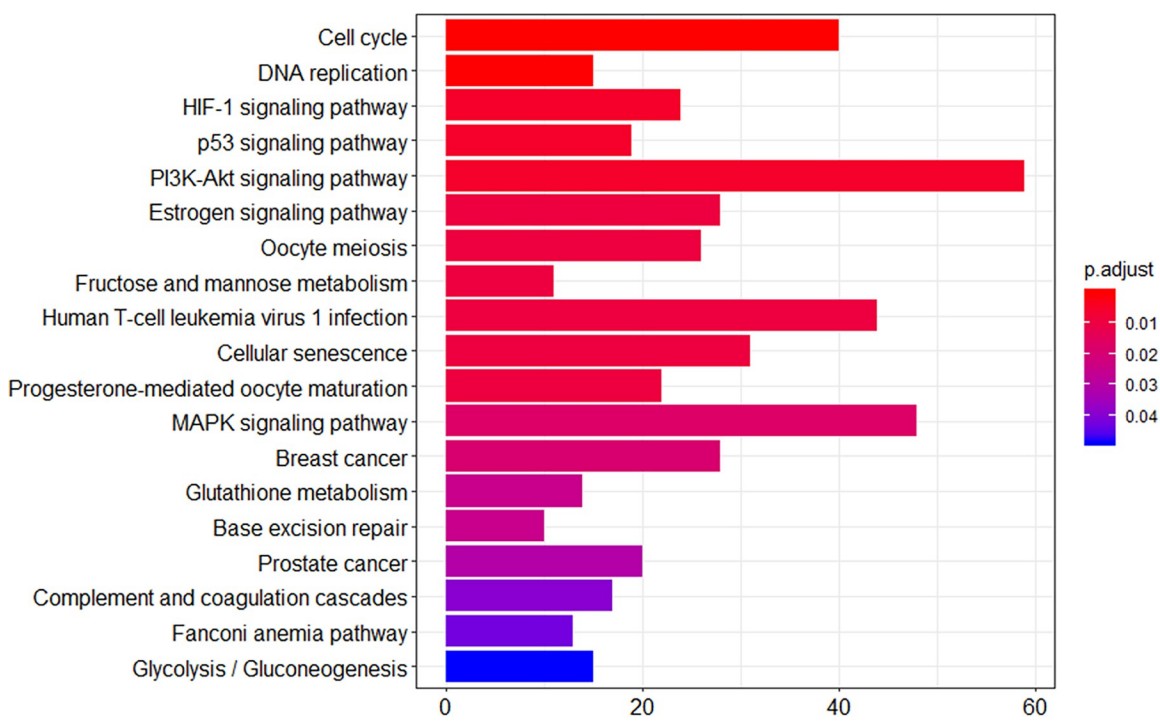

**Fig 9. KEGG pathway analysis results.** The X-axis represents the number of genes. The Y-axis represents pathways.

Notably, we found the hypoxia-inducible factor 1 (HIF1) signaling pathway, glycolysis/gluconeogenesis pathway, and glutathione metabolism among the enriched KEGG pathways. These three pathways are intrinsically related to hypoxia. HIF1 is a master transcriptional regulator of cellular adaptation to hypoxia [40]. Its activity is significantly increased under hypoxic conditions [41]. The glycolytic pathway, also known as the EMP pathway, involves a series of reactions through which glucose and glycogen can be degraded to pyruvate, along with ATP production. The glycolytic pathway can occur under anaerobic and aerobic conditions and is a common mechanism of glucose metabolism in all organisms [42]. Glutathione (GSH) is a small active peptide present in almost every cell type in the body. It plays important antioxidant and detoxification roles [43]. Under normoxic conditions, oxidative metabolism is the main mechanism of energy production that supports cell survival and development [44]. However, under hypoxia, oxidative metabolism is impaired and there

**Table 2. List of differentially expressed genes in MCF7 cells cultured in αMEM vs. DMEM.**

| Symbol | Gene ID | Length (bp) | C1 Expression level | C2 Expression level | log$_2$ ratio (C2/C1) | q-value |
|---|---|---|---|---|---|---|
| P21 | 1026 | 2138 | 21568 | 54319 | 1.32 | 0 |
| GAPDH | 2597 | 1421 | 242199 | 514444 | 1.07 | 0 |
| BIRC3 | 330 | 5174 | 227.6 | 2297.57 | 3.33 | 0 |
| CDK1 | 983 | 1924 | 4011 | 1791 | −1.17 | 7.03E-194 |
| ITGA1 | 3672 | 4811 | 22 | 6 | −1.88 | 0.00099935 |

C1 expression level: Gene expression level in MCF7 cells cultured in DMEM.

C2 expression level: Gene expression level in MCF7 cells cultured in αMEM.

log2FoldChange(C2/C1): The log2 value of ratio of C1-Expression to C2-Expression.

q-value: Adjusted p-value.

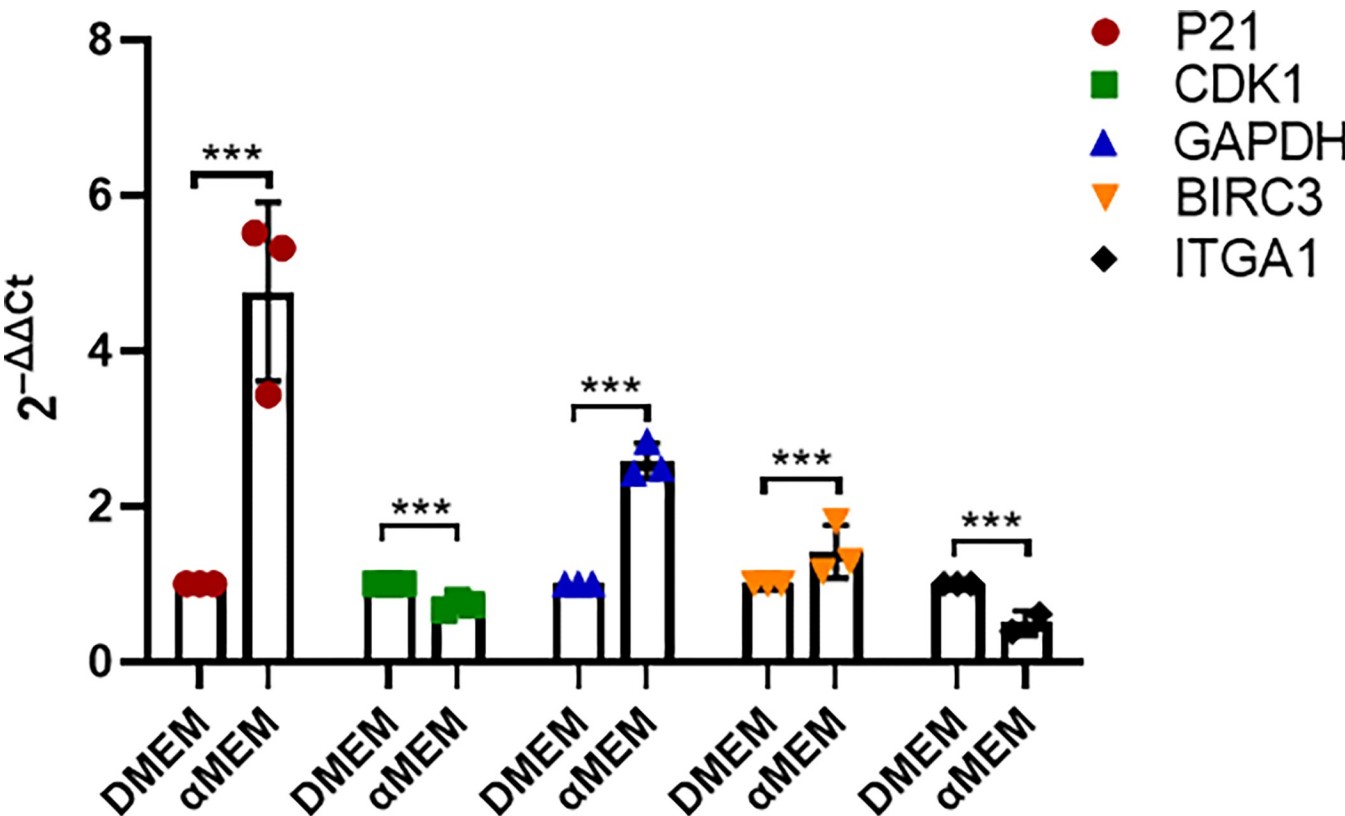

**Fig 10. q-PCR–based validation of the expression patterns of selected genes from the DEGs.** Data are presented as Ct($2^{-\triangle\triangle Ct}$) relative to the control level. Cells were collected after 48 h cultured in DMEM or αMEM. Data are presented as mean ± S.D. of three independent experiments. *P < 0.05, **P < 0.01, ***P < 0.001.

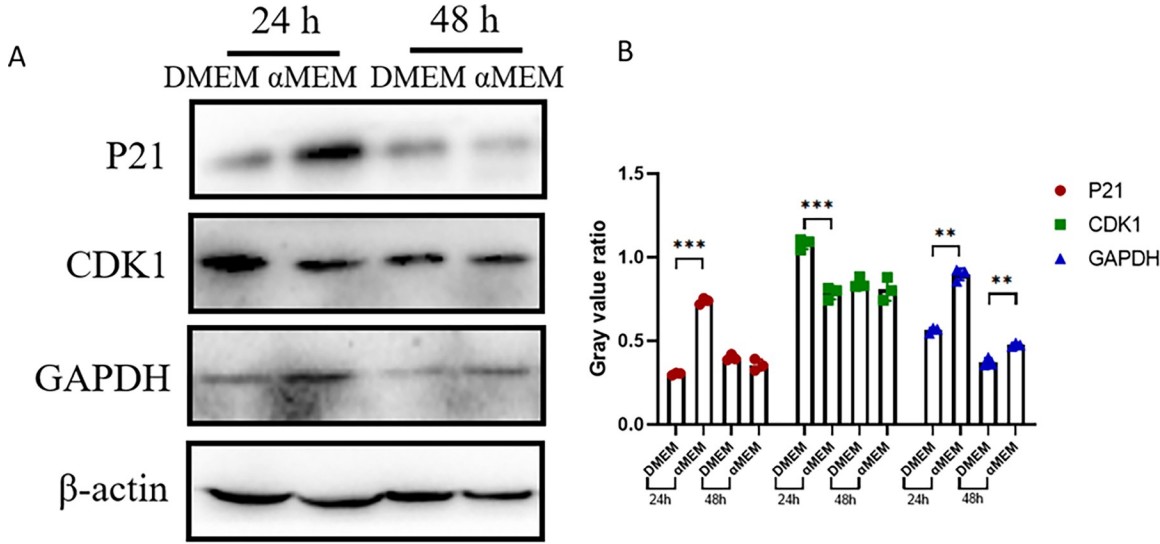

**Fig 11. Assessment of protein levels.** A. Western blot results. B. Quantitation of the results. Cells were collected after 24 h or 48 h cultured in DMEM or αMEM. Data are presented as mean ± S.D. of three independent experiments. Gray value ratio = the gray value of the target band/the gray value of the internal reference band. *P < 0.05, **P < 0.01, ***P < 0.001.

is an accumulation of ROS produced by the electron transport chain. Under such conditions, the HIF1 alpha subunit accumulates and dimerizes with HIF1 beta. This dimer then translocates to the nucleus to activate the transcription of target genes [41], including lactate dehydrogenase (*LDHA*), which is involved in the glycolytic pathway, and pyruvate dehydrogenase kinase 1 (*PDK1*). Activated HIF1 can activate the expression of both *PDK1* and *LDHA* [45] to switch cells from oxidative metabolism to glycolytic metabolism [46,47]. In addition to ATP production, the glycolytic pathway prevents ROS accumulation, which may lead to cellular dysfunction and death. In breast cancer cells, hypoxia induces the expression of genes involved in the glutathione biosynthetic pathway, such as *SLC7A11*, which is a direct target gene of HIF1, and subsequently induces glutathione synthesis [48]. Glutathione is an ROS scavenger that can stoichiometrically reduce NAD(P)H [41]. According to our experiment results, replacement of DMEM (high sugar) with αMEM led to a decrease in ATP production and increase in ROS production. Under hypoxic conditions, the ROS content increases [49]. In our RNA-seq analysis, we found that *PDK1*, *LDHA*, and *SLC7A11* were upregulated in MCF7 cells cultured in αMEM. Combined with the results from the KEGG pathway analysis, we speculate that αMEM culture conditions are hypoxic for MCF7 cells. However, the specific reasons for this remain unclear.

Another notable finding of this study is that the mRNA and protein levels of GAPDH were significantly increased in MCF7 cells cultured in αMEM. GAPDH is a housekeeping gene and its expression levels are considered to remain constant in a given cell type. For this reason, it is widely used as a standard internal reference for gene expression analyses such as qPCR and western blotting. GAPDH has pleiotropic effects on cancer progression, invasiveness, and metastasis, and its levels change under specific circumstances [50]. In MCF7 cells, hypoxia increases the expression of GAPDH both at the transcriptional and translational levels [51], which underscores the need for carefully choosing the internal standard for expression studies.

Although we studied the effects of difference culture media on MCF7 cells, we did not conduct in-depth research on the mechanisms. In addition, we only studied the effects of DMEM and αMEM media on MCF7 cells, the role of more culture media types on different cells should be studied.

## Conclusions

In this study, we compared the effects of Dulbecco's modified Eagle medium (DMEM) and minimum essential medium alpha modification (αMEM) on MCF7 cells. The two media differentially affected the morphology, cell cycle, and proliferation of MCF7 cells, but had no effect on cell death. Replacement of DMEM with αMEM led to a decrease in ATP production and an increase in reactive oxygen species production, but did not affect the cell viability.

RNA-sequencing and bioinformatic analyses revealed 721 significantly upregulated and 1247 downregulated genes in cells cultured in αMEM for 48 h compared with that in cells cultured in DMEM. The enriched GO terms were related to mitosis and cell proliferation. The KEGG analysis revealed cell cycle and DNA replication as the top-two significant pathways that were altered in cells cultured in αMEM. These results show that culture medium exerts a considerable effect on cultured cells. Thus, the stability of the culture system is very important to obtain reliable results warranting the need for optimization of the most suitable medium and its consistent use for cell-based studies.

## Supporting information

**S1 Fig. Correlation analysis between samples.** The X and Y axis represent each sample. The color represents the correlation coefficient (the darker the color, the higher the correlation, the

lighter the color, the lower the correlation).
(PDF)

**S2 Fig. PCA analysis.** X axis represents the contributor rate of first component. Y axis represents the contributor rate of second component. Points represent each sample. The samples in one group shows the same color.
(PDF)

**S1 Table. Clean reads quality metrics.** Total Raw Reads(Mb): The reads amount before filtering. Unit: Mb. Total Clean Reads(Mb): The reads amount after filtering. Unit: Mb. Total Clean Bases(Gb): The total base amount after filtering. Unit: Gb. Clean Reads Q20(%): The Q20 value for the clean reads. Clean Reads Q30(%): The Q20 value for the clean reads. Clean Reads Ratio(%): The ratio of the amount of clean reads.
(PDF)

**S1 Raw images.**
(PDF)

## Author Contributions

**Conceptualization:** Yang Jiao, Lin Lu, Bingrong Zheng.

**Formal analysis:** Hongbo Zhao.

**Funding acquisition:** Lin Lu, Bingrong Zheng.

**Investigation:** Lin Lu, Xiangyu Zhao, Yanchun Wang.

**Writing – original draft:** Yang Jiao.

**Writing – review & editing:** Hongbo Zhao, Bingrong Zheng.

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
