## [Decision Letter · Decision Letter 0]

22 Nov 2023

PONE-D-23-29345Transcriptome-wide analysis of the difference between MCF7 cells cultured in DMEM or αMEMPLOS ONE

Dear Dr. Zheng,

Thank you for submitting your manuscript to PLOS ONE. After careful consideration, we feel that it has merit but does not fully meet PLOS ONE’s publication criteria as it currently stands. Therefore, we invite you to submit a revised version of the manuscript that addresses the points raised during the review process.

We look forward to receiving your revised manuscript.

Kind regards,

Sadiq Umar

Academic Editor

PLOS ONE

Journal Requirements:

https://www.hindawi.com/journals/sci/2018/5912194/

In your revision ensure you cite all your sources (including your own works), and quote or rephrase any duplicated text outside the methods section. Further consideration is dependent on these concerns being addressed.

"This work was supported by Major Project of Yunnan Science and Technology Program [grant number 202002AA100007 and 202102AA100007-3], Scientific Research Foundation of the Education Department of Yunnan Province [grant number 2021J0226], Joint Special Funds for the Department of Science and Technology of Yunnan Province-Kunming Medical University [grant numbers 2019FE001(-176), 202101AY070001-075]."

6. Thank you for stating the following in your Competing Interests section:  

'The authors declare no conflicts of interest."

7. PLOS ONE now requires that authors provide the original uncropped and unadjusted images underlying all blot or gel results reported in a submission’s figures or Supporting Information files. This policy and the journal’s other requirements for blot/gel reporting and figure preparation are described in detail at https://journals.plos.org/plosone/s/figures#loc-blot-and-gel-reporting-requirements and https://journals.plos.org/plosone/s/figures#loc-preparing-figures-from-image-files. When you submit your revised manuscript, please ensure that your figures adhere fully to these guidelines and provide the original underlying images for all blot or gel data reported in your submission. See the following link for instructions on providing the original image data: https://journals.plos.org/plosone/s/figures#loc-original-images-for-blots-and-gels. 

Reviewers' comments:

Reviewer's Responses to Questions

**Comments to the Author**

1. Is the manuscript technically sound, and do the data support the conclusions?

Reviewer #1: No

Reviewer #2: Yes

2. Has the statistical analysis been performed appropriately and rigorously? 

Reviewer #1: No

Reviewer #2: Yes

3. Have the authors made all data underlying the findings in their manuscript fully available?

Reviewer #1: No

Reviewer #2: Yes

4. Is the manuscript presented in an intelligible fashion and written in standard English?

Reviewer #1: No

Reviewer #2: Yes

5. Review Comments to the Author

Reviewer #1: The manuscript entitled “Transcriptome-wide analysis of the difference between MCF7 cells 2 cultured in DMEM or αMEM” aims to demonstrate the effect of culture medium on MCF7 cells in terms of transcriptomic profiling. The manuscript is poorly written and lack the significance and novelty of the study. My comments are as follows.

1. What is the rationale of this study? Cells grown in different culture media will definitely give different phenotype. Every cell culture has its definite and defined media to culture and proceed the analysis.

2. Pease provide the details of RNA sequencing, like library preparation and sequencing. Please provide PCA for the QC of RNA sequencing.

3. English mistakes are prevalent throughout the manuscript. Please rectify.

4. Missing legends of the figures. Hard to assess the figures and correlate with the results.

5. The manuscript lacks mechanism.

Reviewer #2: Title: Transcriptome-wide analysis of the difference between MCF7 cells cultured in DMEM or αMEM

Reviewer’s report

The authors have examined the effect of two different types of cell culture media on breast cancer model 'MCF7'. Based on RNA sequencing data they found that culturing the MCF7 cells in different culture media led to differential gene expression. Moreover, they found that cells that were cultured in alpha-MEM were hypoxic. This study reveals that culture media plays an important role on cell growth property at morphological, biological and transcriptional level, thus it is utmost important to choose the correct cell culture media for a specific cell type to obtain reliable results. The present study reveals interesting findings; however, there are several essential points that need to be addressed before publication.

Minor Comments:

1) In Introduction line no 75, authors have mentioned about NP cells without its introduction. Please explain a little bit more about NP cells and its significance here in this current study.

2) It would be interesting if authors could also show the effect of different culture media on cell growth and proliferation by colony forming assay.

3) In all the results (graphs) please include individual data points that would give a clear indication about number of replicates in each group.

4) What was the effect of different cultures media on cell attachment to the surface of the cell culture flasks. Were there any differences observed by authors in terms of cell attachment.

5) Did authors find any differential effect of culturing MCF7 in Mitochondrial respiration and intracellular ATP levels or morphological changes at organelle level?

6) It would be interesting to look at the effect of culturing MCF7 in different culture media on the expression of MCF7 specific genes like Insulin like growth factor binding protein 2, WNT7B and LHR expression?

6. PLOS authors have the option to publish the peer review history of their article (what does this mean?). If published, this will include your full peer review and any attached files.

Reviewer #1: No

Reviewer #2: No

---

## [Author Response · Author response to Decision Letter 0]

21 Jan 2024

Dear Editor,

Thank you for giving us the opportunity to revise our manuscript entitled “Transcriptome-wide analysis of the difference between MCF7 cells cultured in DMEM or αMEM (Manuscript ID: PONE-D-23-29345)". Sorry for the late submission. We now have completed our revision. The responds to the reviewer’s comments are as flowing:

We have revised the format of our article according to PLOS ONE's style requirements, including file naming.

2. We noticed you have some minor occurrence of overlapping text with the following previous publication(s), which needs to be addressed: https://www.hindawi.com/journals/sci/2018/5912194/

In your revision ensure you cite all your sources (including your own works), and quote or rephrase any duplicated text outside the methods section. Further consideration is dependent on these concerns being addressed.

We have quoted or rephrased the duplicated text outside the methods section.

The mRNA-seq data from this study has been deposited in the NCBI sequence read archive under the BioProject number PRJNA779251, with the fq files spanning accession numbers SRR16914480-SRR16914481.

We have corrected the ‘Funding Information’ according to the ‘Financial Disclosure’ sections, and we ensure that the grant numbers are correct.

"This work was supported by Major Project of Yunnan Science and Technology Program [grant number 202002AA100007 and 202102AA100007-3], Scientific Research Foundation of the Education Department of Yunnan Province [grant number 2021J0226], Joint Special Funds for the Department of Science and Technology of Yunnan Province-Kunming Medical University [grant numbers 2019FE001(-176), 202101AY070001-075]."

We have corrected the ‘Funding Information’ according to the ‘Financial Disclosure’ sections, and we ensure that the grant numbers are correct, and we have added it in the cover letter.

6. Thank you for stating the following in your Competing Interests section: 

'The authors declare no conflicts of interest."

We have completed our Competing Interests on the online submission form and cover letter.

7. PLOS ONE now requires that authors provide the original uncropped and unadjusted images underlying all blot or gel results reported in a submission’s figures or Supporting Information files. This policy and the journal’s other requirements for blot/gel reporting and figure preparation are described in detail at https://journals.plos.org/plosone/s/figures#loc-blot-and-gel-reporting-requirements and https://journals.plos.org/plosone/s/figures#loc-preparing-figures-from-image-files. When you submit your revised manuscript, please ensure that your figures adhere fully to these guidelines and provide the original underlying images for all blot or gel data reported in your submission. See the following link for instructions on providing the original image data: https://journals.plos.org/plosone/s/figures#loc-original-images-for-blots-and-gels.

We have uploaded the original uncropped and unadjusted images underlying all blot results reported in a submission’s figures as “S1_raw_images”.

Response to Reviewer 1

Thank you for your careful reading and giving us useful suggestions.

The manuscript entitled “Transcriptome-wide analysis of the difference between MCF7 cells 2 cultured in DMEM or αMEM” aims to demonstrate the effect of culture medium on MCF7 cells in terms of transcriptomic profiling. The manuscript is poorly written and lack the significance and novelty of the study.

We have supplemented the experiments based on the comments from reviewers and made revisions to our manuscript to highlight the key points. We hope this manuscript is appropriate for publication.

1. What is the rationale of this study? Cells grown in different culture media will definitely give different phenotype. Every cell culture has its definite and defined media to culture and proceed the analysis.

Although it is well known that different culture media can cause different phenotypes in cells, we do not yet know in which specific aspects they impact on, especially on the cellular genome. Also, few people would spend time and experience doing this study. Therefore, our research can provide a reference.

2. Pease provide the details of RNA sequencing, like library preparation and sequencing. Please provide PCA for the QC of RNA sequencing.

We have added these contents to “RNA isolation and RNA-Seq library preparation” and “Differential gene expression and gene enrichment analysis” of “Materials and Methods” in manuscript.

3. English mistakes are prevalent throughout the manuscript. Please rectify.

Thank you for your reminder, and we have rectified these mistakes.

4. Missing legends of the figures. Hard to assess the figures and correlate with the results.

We have added the legends of the figures.

5. The manuscript lacks mechanism.

We will further study the mechanism in the future.

Response to Reviewer 2

Thank you for your careful reading and giving us useful suggestions.

The authors have examined the effect of two different types of cell culture media on breast cancer model 'MCF7'. Based on RNA sequencing data they found that culturing the MCF7 cells in different culture media led to differential gene expression. Moreover, they found that cells that were cultured in alpha-MEM were hypoxic. This study reveals that culture media plays an important role on cell growth property at morphological, biological and transcriptional level, thus it is utmost important to choose the correct cell culture media for a specific cell type to obtain reliable results. The present study reveals interesting findings; however, there are several essential points that need to be addressed before publication.

Minor Comments:

1) In Introduction line no 75, authors have mentioned about NP cells without its introduction. Please explain a little bit more about NP cells and its significance here in this current study.

Thank you for your reminder. We have carefully read the context and found that NP cells are indeed abrupt and unimportant here in this current study, so we have deleted this sentence.

2) It would be interesting if authors could also show the effect of different culture media on cell growth and proliferation by colony forming assay.

We have performed cell colony forming experiments and included the results in the manuscript.

3) In all the results (graphs) please include individual data points that would give a clear indication about number of replicates in each group.

We have remade the graphs to include individual data points.

4) What was the effect of different cultures media on cell attachment to the surface of the cell culture flasks. Were there any differences observed by authors in terms of cell attachment.

We have performed cell attachment experiments and included the results in the manuscript.

5) Did authors find any differential effect of culturing MCF7 in Mitochondrial respiration and intracellular ATP levels or morphological changes at organelle level?

We have detected the intracellular ATP levels and included the results in the manuscript.

6) It would be interesting to look at the effect of culturing MCF7 in different culture media on the expression of MCF7 specific genes like Insulin like growth factor binding protein 2, WNT7B and LHR expression?

We conducted t-tests on the expression values (FPKM) of three genes. The results were that the p-value of the Insulin like growth factor binding protein 2 gene was 0.01977, which was slightly higher in C1, and the p-value of WNT7B and LHR were 0.05227 and 0.9694, which were no significant difference.

---

## [Editor Report · Decision Letter 1]

23 Jan 2024

Transcriptome-wide analysis of the difference between MCF7 cells cultured in DMEM or αMEM

PONE-D-23-29345R1

Dear Dr. Zheng,

We’re pleased to inform you that your manuscript has been judged scientifically suitable for publication and will be formally accepted for publication once it meets all outstanding technical requirements.

Kind regards,

Sadiq Umar

Academic Editor

PLOS ONE

---

## [Editor Report · Acceptance letter]

20 Mar 2024

PONE-D-23-29345R1 

PLOS ONE

Dear Dr. Zheng, 

I'm pleased to inform you that your manuscript has been deemed suitable for publication in PLOS ONE. Congratulations! Your manuscript is now being handed over to our production team.

Kind regards, 

on behalf of

Dr. Sadiq Umar 

Academic Editor

PLOS ONE